# NBDT: NEURAL-BACKED DECISION TREE

**Alvin Wan**$_1$, **Lisa Dunlap**$^*_1$, **Daniel Ho**$^*$, **Jihan Yin**$_1$, **Scott Lee**$_1$, **Suzanne Petryk**$_1$,
**Sarah Adel Bargal**$_2$, **Joseph E. Gonzalez**$_1$
UC Berkeley$_1$, Boston University$_2$

{alvinwan,ldunlap,danielho,jihan_yin,scott.lee.3898,spetryk,jegonzal}@berkeley.edu

sbargal@bu.edu

## ABSTRACT

Machine learning applications such as finance and medicine demand accurate and justifiable predictions, barring most deep learning methods from use. In response, previous work combines decision trees with deep learning, yielding models that (1) sacrifice interpretability for accuracy or (2) sacrifice accuracy for interpretability. We forgo this dilemma by *jointly improving* accuracy and interpretability using Neural-Backed Decision Trees (NBDTs). NBDTs replace a neural network's final linear layer with a differentiable sequence of decisions and a surrogate loss. This forces the model to learn high-level concepts and lessens reliance on highly-uncertain decisions, yielding (1) accuracy: NBDTs match or outperform modern neural networks on CIFAR, ImageNet and better generalize to unseen classes by up to 16%. Furthermore, our surrogate loss improves the *original* model's accuracy by up to 2%. NBDTs also afford (2) interpretability: improving human trust by clearly identifying model mistakes and assisting in dataset debugging. Code and pretrained NBDTs are at github.com/alvinwan/neural-backed-decision-trees.

## 1 INTRODUCTION

Many computer vision applications (e.g. medical imaging and autonomous driving) require insight into the model's decision process, complicating applications of deep learning which are traditionally black box. Recent efforts in explainable computer vision attempt to address this need and can be grouped into one of two categories: (1) saliency maps and (2) sequential decision processes. Saliency maps retroactively explain model predictions by identifying which pixels most affected the prediction. However, by focusing on the input, saliency maps fail to capture the model's decision making process. For example, saliency offers no insight for a misclassification when the model is "looking" at the right object for the wrong reasons. Alternatively, we can gain insight into the model's decision process by breaking up predictions into a sequence of smaller semantically meaningful decisions as in rule-based models like decision trees. However, existing efforts to fuse deep learning and decision trees suffer from (1) significant accuracy loss, relative to contemporary models (e.g., residual networks), (2) reduced interpretability due to accuracy optimizations (e.g., impure leaves and ensembles), and (3) tree structures that offer limited insight into the model's credibility.

To address these, we propose **Neural-Backed Decision Trees (NBDTs)** to jointly improve *both* (1) accuracy and (2) interpretability of modern neural networks, utilizing decision rules that preserve (3) properties like sequential, discrete decisions; pure leaves; and non-ensembled predictions. These properties in unison enable unique insights, as we show. We acknowledge that there is no universally-accepted definition of interpretability (Lundberg et al., 2020; Doshi-Velez & Kim, 2017; Lipton, 2016), so to show interpretability, we adopt a definition offered by Poursabzi-Sangdeh et al. (2018): A model is interpretable if a human can validate its prediction, determining when the model has made a sizable mistake. We picked this definition for its importance to downstream benefits we can evaluate, specifically (1) model or dataset debugging and (2) improving human trust. To accomplish this, NBDTs replace the final linear layer of a neural network with a differentiable oblique decision tree and, unlike its predecessors (*i.e.* decision trees, hierarchical classifiers), uses a hierarchy derived from model parameters, does not employ a hierarchical softmax, and can be created from *any* existing classification neural network without architectural modifications. These improvements

---

$^*$denotes equal contribution

tailor the hierarchy to the network rather than overfit to the feature space, lessens the decision tree's reliance on highly uncertain decisions, and encourages accurate recognition of high-level concepts. These benefits culminate in joint improvement of accuracy and interpretability. Our contributions:

1. We propose a *tree supervision loss*, yielding NBDTs that match/outperform and out-generalize modern neural networks (WideResNet, EfficientNet) on ImageNet, TinyImageNet200, and CIFAR100. Our loss also improves the *original* model by up to 2%.

2. We propose alternative hierarchies for oblique decision trees – *induced hierarchies* built using pre-trained neural network weights – that outperform both data-based hierarchies (e.g. built with information gain) and existing hierarchies (e.g. WordNet), in accuracy.

3. We show NBDT explanations are more helpful to the user when identifying model mistakes, preferred when using the model to assist in challenging classification tasks, and can be used to identify ambiguous ImageNet labels.

## 2 RELATED WORKS

**Saliency Maps.** Numerous efforts (Springenberg et al., 2014; Zeiler & Fergus, 2014; Simonyan et al., 2013; Zhang et al., 2016; Selvaraju et al., 2017; Ribeiro et al., 2016; Petsiuk et al., 2018; Sundararajan et al., 2017) have explored the design of saliency maps identifying pixels that most influenced the model's prediction. White-box techniques (Springenberg et al., 2014; Zeiler & Fergus, 2014; Simonyan et al., 2013; Selvaraju et al., 2017; Sundararajan et al., 2017) use the network's parameters to determine salient image regions, and black-box techniques (Ribeiro et al., 2016; Petsiuk et al., 2018) determine pixel importance by measuring the prediction's response to perturbed inputs. However, saliency does not explain the model's decision process (e.g. Was the model confused early on, distinguishing between *Animal* and *Vehicle*? Or is it only confused between dog breeds?).

**Transfer to Explainable Models.** Prior to the recent success of deep learning, decision trees were state-of-the-art on a wide variety of learning tasks and the gold standard for interpretability. Despite this recency, study at the intersection of neural network and decision tree dates back three decades, where neural networks were seeded with decision tree weights (Banerjee, 1990; 1994; Ivanova & Kubat, 1995a;b), and decision trees were created from neural network queries (Krishnan et al., 1999; Boz, 2000; Dancey et al., 2004; Craven & Shavlik, 1996; 1994), like distillation (Hinton et al., 2015). The modern analog of both sets of work (Humbird et al., 2018; Siu, 2019; Frosst & Hinton, 2017) evaluate on feature-sparse, sample-sparse regimes such as the UCI datasets (Dua & Graff, 2017) or MNIST (LeCun et al., 2010) and *perform poorly* on standard image classification tasks.

**Hybrid Models.** Recent work produces hybrid decision tree and neural network models to scale up to datasets like CIFAR10 (Krizhevsky, 2009), CIFAR100 (Krizhevsky, 2009), TinyImageNet (Le & Yang, 2015), and ImageNet (Deng et al., 2009). One category of models organizes the neural network into a hierarchy, dynamically selecting branches to run inference (Veit & Belongie, 2018; McGill & Perona, 2017; Teja Mullapudi et al., 2018; Redmon & Farhadi, 2017; Murdock et al., 2016). However, these models use *impure leaves* resulting in uninterpretatble, stochastic paths. Other approaches fuse deep learning into each decision tree node: an entire neural network (Murthy et al., 2016), several layers (Murdock et al., 2016; Roy & Todorovic, 2016), a linear layer (Ahmed et al., 2016), or some other parameterization of neural network output (Kontschieder et al., 2015). These models see reduced interpretability by using k-way decisions with large k (via depth-2 trees) (Ahmed et al., 2016; Guo et al., 2018) or employing an ensemble (Kontschieder et al., 2015; Ahmed et al., 2016), which is often referred to as a "black box" (Carvalho et al., 2019; Rudin, 2018).

**Hierarchical Classification** (Silla & Freitas, 2011). One set of approaches directly uses a pre-existing hierarchy over classes, such as WordNet (Redmon & Farhadi, 2017; Brust & Denzler, 2019; Deng et al.). However *conceptual similarity is not indicative of visual similarity*. Other models build a hierarchy using the training set directly, via a classic data-dependent metric like Gini impurity (Alaniz & Akata, 2019) or information gain (Rota Bulo & Kontschieder, 2014; Biçici et al., 2018). These models are instead *prone to overfitting*, per (Tanno et al., 2019). Finally, several works introduce hierarchical surrogate losses (Wu et al., 2017; Deng et al., 2012), such as hierarchical softmax (Mohammed & Umaashankar, 2018), but as the authors note, these methods quickly suffer from major accuracy loss with more classes or higher-resolution images (e.g. beyond CIFAR10). We demonstrate hierarchical classifiers attain higher accuracy *without* a hierarchical softmax.

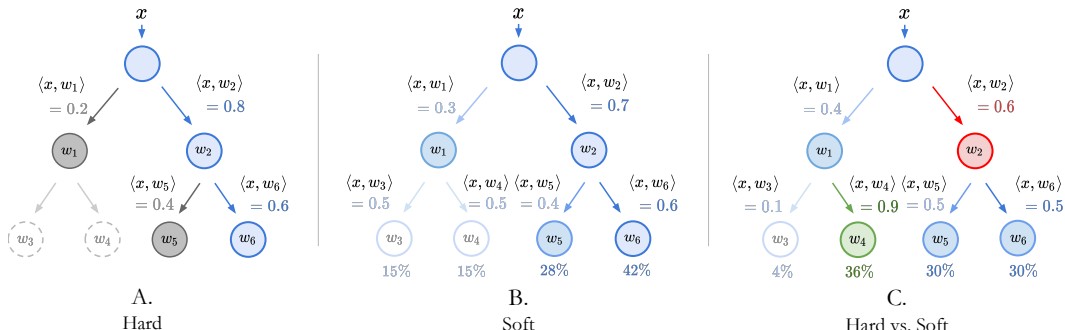

**Figure 1: Hard and Soft Decision Trees. A. Hard:** is the classic "hard" oblique decision tree. Each node picks the child node with the largest inner product, and visits that node next. Continue until a leaf is reached. **B. Soft:** is the "soft" variant, where each node simply returns probabilities, as normalized inner products, of each child. For each leaf, compute the probability of its path to the root. Pick leaf with the highest probability. **C. Hard vs. Soft:** Assume $w_4$ is the correct class. With hard inference, the mistake at the root (red) is irrecoverable. However, with soft inference, the highly-uncertain decisions at the root and at $w_2$ are superseded by the highly certain decision at $w_3$ (green). This means the model can still correctly pick $w_4$ despite a mistake at the root. In short, soft inference can tolerate mistakes in highly uncertain decisions.

## 3 METHOD

Neural-Backed Decision Trees (NBDTs) replace a network's final linear layer with a decision tree. Unlike classical decision trees or many hierarchical classifiers, NBDTs use path probabilities for inference (Sec 3.1) to tolerate highly-uncertain intermediate decisions, build a hierarchy from pre-trained model weights (Sec 3.2 & 3.3) to lessen overfitting, and train with a hierarchical loss (Sec 3.4) to significantly better learn high-level decisions (e.g., *Animal* vs. *Vehicle*).

### 3.1 INFERENCE

Our NBDT first featurizes each sample using the neural network backbone; the backbone consists of all neural network layers before the final linear layer. Second, we run the final fully-connected layer as an oblique decision tree. However, (a) a classic decision tree cannot recover from a mistake early in the hierarchy and (b) just running a classic decision tree on neural features drops accuracy significantly, by up to 11% (Table 2). Thus, we present modified decision rules (Figure 1, B):

**1. Seed oblique decision rule weights with neural network weights.** An oblique decision tree supports only binary decisions, using a hyperplane for each decision. Instead, we associate a weight vector $n_i$ with each node. For leaf nodes, where $i = k \in [1, K]$, each $n_i = w_k$ is a row vector from the fully-connected layer's weights $W \in \mathbb{R}^{D \times K}$. For all inner nodes, where $i \in [K + 1, N]$, find all leaves $k \in L(i)$ in node $i$'s subtree and average their weights: $n_i = \sum_{k \in L(i)} w_k / |L(i)|$.

**2. Compute node probabilities.** Child probabilities are given by softmax inner products. For each sample $x$ and node $i$, compute the probability of each child $j \in C(i)$ using $p(j|i) = \text{SOFTMAX}(\langle \vec{n}_i, x \rangle)[j]$, where $\vec{n}_i = (\langle n_j, x \rangle)_{j \in C(i)}$.

**3. Pick a leaf using path probabilities.** Inspired by Deng et al. (2012), consider a leaf, its class $k$ and its path from the root $P_k$. The probability of each node $i \in P_k$ traversing the next node in the path $C_k(i) \in P_k \cap C(i)$ is denoted $p(C_k(i)|i)$. Then, the probability of leaf and its class $k$ is

$$p(k) = \Pi_{i \in P_k} p(C_k(i)|i) \tag{1}$$

In soft inference, the final class prediction $\hat{k}$ is defined over these class probabilities,

$$\hat{k} = \text{argmax}_k p(k) = \text{argmax}_k \Pi_{i \in P_k} p(C_k(i)|i) \tag{2}$$

Our inference strategy has two benefits: (a) Since the architecture is unchanged, the fully-connected layer can be run regularly (Table 5) or as decision rules (Table 1), and (b) unlike decision trees and

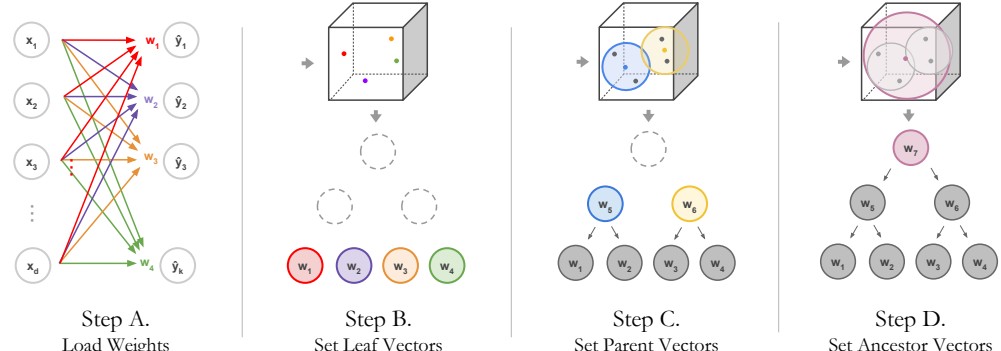

Figure 2: **Building Induced Hierarchies. Step A.** Load the weights of a pre-trained model's final fully-connected layer, with weight matrix $W \in \mathbb{R}^{D \times K}$. **Step B.** Take rows $w_k \in W$ and normalize for each leaf node's weight. For example, the red $w_1$ in A is assigned to the red leaf in B. **Step C.** Average each pair of leaf nodes for the parents' weight. For example, $w_1$ and $w_2$ (red and purple) in B are averaged to make $w_5$ (blue) in C. **Step D.** For each ancestor, average all leaf node weights in its subtree. That average is the ancestor's weight. Here, the ancestor *is* the root, so its weight is the average of all leaf weights $w_1, w_2, w_3, w_4$.

other conditionally-executed models (Tanno et al., 2019; Veit & Belongie, 2018), our method can recover from a mistake early in the hierarchy with sufficient uncertainty in the incorrect path (Figure 1 C, Appendix Table 7). This inference mode bests classic tree inference (Appendix C.2).

## 3.2 Building Induced Hierarchies

Existing decision-tree-based methods use (a) hierarchies built with data-dependent heuristics like information gain or (b) existing hierarchies like WordNet. However, the former overfits to the data, and the latter focuses on conceptual rather than visual similarity: For example, by virtue of being an animal, *Bird* is closer to *Cat* than to *Plane*, according to WordNet. However, the opposite is true for visual similarity: by virtue of being in the sky, *Bird* is more visually similar to *Plane* than to *Cat*. Thus, to prevent overfitting and reflect visual similarity, we build a hierarchy using model weights.

Our hierarchy requires pre-trained model weights. Take row vectors $w_k : k \in [1, K]$, each representing a class, from the fully-connected layer weights $W$. Then, run hierarchical agglomerative clustering on the normalized class representatives $w_k / \|w_k\|_2$. Agglomerative clustering decides which nodes and groups of nodes are iteratively paired. As described in Sec 3.1, each leaf node's weight is a row vector $w_k \in W$ (Figure 2, Step B) and each inner node's weight $n_i$ is the average of its leaf node's weights (Figure 2, Step C). This hierarchy is the *induced hierarchy* (Figure 2).

## 3.3 Labeling Decision Nodes with WordNet

WordNet is a hierarchy of nouns. To assign WordNet meaning to nodes, we compute the earliest common ancestor for all leaves in a subtree: For example, say *Dog* and *Cat* are two leaves that share a parent. To find WordNet meaning for the parent, find all ancestor concepts that *Dog* and *Cat* share, like *Mammal*, *Animal*, and *Living Thing*. The earliest shared ancestor is *Mammal*, so we assign *Mammal* to the parent of *Dog* and *Cat*. We repeat for all inner nodes.

However, the WordNet corpus is lacking in concepts that are not themselves objects, like object attributes (e.g., *Pencil* and *Wire* are both cylindrical) and (b) abstract visual ideas like context (e.g., *fish* and *boat* are both aquatic). Many of these which are littered across our induced hierarchies (Appendix Figure 14). Despite this limitation, we use WordNet to assign meaning to intermediate decision nodes, with more sophisticated methods left to future work.

## 3.4 Fine-tuning with Tree Supervision Loss

Even though standard cross entropy loss separates representatives for each leaf, it is not trained to separate representatives for each inner node (Table 3, "None"). To amend this, we add a *tree super-*

**Table 1: Results.** NBDT outperforms competing decision-tree-based methods by up to 18% and can also outperform the *original* neural network by $\sim$ 1%. "Expl?" indicates the method retains interpretable properties: pure leaves, sequential decisions, non-ensemble. Methods without this check see reduced interpretability. We bold the highest decision-tree-based accuracy. These results are taken directly from the original papers (*n/a* denotes results missing from original papers): XOC (Alaniz & Akata, 2019), DCDJ (Baek et al., 2017), NofE (Ahmed et al., 2016), DDN (Murthy et al., 2016), ANT (Tanno et al., 2019), CNN-RNN (Guo et al., 2018). We train DNDF (Kontschieder et al., 2015) with an updated R18 backbone, as they did not report CIFAR accuracy.

| Method | Backbone | Expl? | CIFAR10 | CIFAR100 | TinyImageNet |
|--------|----------|-------|---------|----------|--------------|
| NN | WideResNet28x10 | ✗ | 97.62% | 82.09% | 67.65% |
| ANT-A* | *n/a* | ✓ | 93.28% | *n/a* | *n/a* |
| DDN | NiN | ✗ | 90.32% | 68.35% | *n/a* |
| DCDJ | NiN | ✗ | *n/a* | 69.0% | *n/a* |
| NofE | ResNet56-4x | ✗ | *n/a* | 76.24% | *n/a* |
| CNN-RNN | WideResNet28x10 | ✓ | *n/a* | 76.23% | *n/a* |
| NBDT-S (Ours) | WideResNet28x10 | ✓ | **97.55%** | **82.97%** | **67.72%** |
| NN | ResNet18 | ✗ | 94.97% | 75.92% | 64.13% |
| DNDF | ResNet18 | ✗ | 94.32% | 67.18% | 44.56% |
| XOC | ResNet18 | ✓ | 93.12% | *n/a* | *n/a* |
| DT | ResNet18 | ✓ | 93.97% | 64.45% | 52.09% |
| NBDT-S (Ours) | ResNet18 | ✓ | **94.82%** | **77.09%** | **64.23%** |

*vision loss*, a cross entropy loss over the class distribution of path probabilities $\mathcal{D}_{\text{nbdt}} = \{p(k)\}_{k=1}^{K}$ (Eq. 1) from Sec 3.1, with time-varying weights $\omega_t, \beta_t$ where $t$ is the epoch count:

$$\mathcal{L} = \beta_t \underbrace{\text{CROSSENTROPY}(\mathcal{D}_{\text{pred}}, \mathcal{D}_{\text{label}})}_{\mathcal{L}_{\text{original}}} + \omega_t \underbrace{\text{CROSSENTROPY}(\mathcal{D}_{\text{nbdt}}, \mathcal{D}_{\text{label}})}_{\mathcal{L}_{\text{soft}}} \quad (3)$$

Our tree supervision loss $\mathcal{L}_{\text{soft}}$ requires a pre-defined hierarchy. We find that (a) tree supervision loss damages learning speed early in training, when leaf weights are nonsensical. Thus, our tree supervision weight $\omega_t$ grows linearly from $\omega_0 = 0$ to $\omega_T = 0.5$ for CIFAR10, CIFAR100, and to $\omega_T = 5$ for TinyImageNet, ImageNet; $\beta_t \in [0, 1]$ decays linearly over time. (b) We re-train where possible, fine-tuning with $\mathcal{L}_{\text{soft}}$ only when the original model accuracy is not reproducible. (c) Unlike hierarchical softmax, our path-probability cross entropy loss $\mathcal{L}_{\text{soft}}$ disproportionately up-weights decisions earlier in the hierarchy, encouraging accurate high-level decisions; this is reflected our out-generalization of the baseline neural network by up to 16% to unseen classes (Table 6).

## 4 EXPERIMENTS

NBDTs obtain state-of-the-art results for interpretable models and match or outperform modern neural networks on image classification. We report results on different models (ResNet, WideResNet, EfficientNet) and datasets (CIFAR10, CIFAR100, TinyImageNet, ImageNet). We additionally conduct ablation studies to verify the hierarchy and loss designs, find that our training procedure improves the *original* neural network's accuracy by up to 2%, and show that NBDTs improve generalization to unseen classes by up to 16%. All reported improvements are absolute.

### 4.1 RESULTS

**Small-scale Datasets.** Our method (Table 1) matches or outperforms recently state-of-the-art neural networks. On CIFAR10 and TinyImageNet, NBDT accuracy falls within 0.15% of the baseline neural network. On CIFAR100, NBDT accuracy outperforms the baseline by $\sim$1%.

**Large-scale Dataset.** On ImageNet (Table 3), NBDTs obtain 76.60% top-1 accuracy, outperforming the strongest competitor NofE by 15%. Note that we take the best competing results for any decision-tree-based method, but the strongest competitors hinder interpretability by using ensembles of models like a decision forest (DNDF, DCDJ) or feature shallow trees with only depth 2 (NofE).

**Figure 3: ImageNet Results.** NBDT outperforms all competing decision-tree-based methods by at least 14%, staying within 0.6% of EfficientNet accuracy. "EfficientNet" is EfficientNet-EdgeTPU-Small.

| Method | NBDT (ours) | NBDT (ours) | XOC | NofE |
|---|---|---|---|---|
| Backbone | EfficientNet | ResNet18 | ResNet152 | AlexNet |
| Original Acc | 77.23% | 60.76% | 78.31% | 56.55% |
| Delta Acc | -0.63% | +0.50% | -17.5% | +4.7% |
| Explainable Acc | **76.60%** | 61.26% | 60.77% | 61.29% |

**Table 2: Comparisons of Hierarchies.** We demonstrate that our weight-space hierarchy bests taxonomy and data-dependent hierarchies. In particular, the induced hierarchy achieves better performance than (a) the WordNet hierarchy, (b) a classic decision tree's information gain hierarchy, built over neural features ("Info Gain"), and (c) an oblique decision tree built over neural features ("OC1").

| Dataset | Backbone | Original | Induced | Info Gain | WordNet | OC1 |
|---|---|---|---|---|---|---|
| CIFAR10 | ResNet18 | *94.97%* | **94.82%** | 93.97% | 94.37% | 94.33% |
| CIFAR100 | ResNet18 | *75.92%* | **77.09%** | 64.45% | 74.08% | 38.67% |
| TinyImageNet200 | ResNet18 | *64.13%* | **64.23%** | 52.09% | 60.26% | 15.63% |

### 4.2 ANALYSIS

Analyses show that our NBDT improvements are dominated by significantly improved ability to distinguish higher-level concepts (e.g., *Animal* vs. *Vehicle*).

**Comparison of Hierarchies.** Table 2 shows that our induced hierarchies outperform alternatives. In particular, *data-dependent* hierarchies overfit, and the existing *WordNet hierarchy* focuses on conceptual rather than visual similarity.

**Comparisons of Losses.** Previous work suggests hierarchical softmax (Appendix C.1) is necessary for hierarchical classifiers. However, our results suggest otherwise: NBDTs trained with hierarchical softmax see ∼3% less accuracy than with tree supervision loss on TinyImageNet (Table 3).

**Original Neural Network.** Per Sec 3.1, we can run the original neural network's fully-connected layer normally, after training with tree supervision loss. Using this, we find that the original neural network's accuracy improves by up to 2% on CIFAR100, TinyImageNet (Table 5).

**Zero-Shot Superclass Generalization.** We define a "superclass" to be the hypernym of several classes. (e.g. *Animal* is a superclass of *Cat* and *Dog*). Using WordNet (per Sec 3.2), we (1) identify which superclasses each NBDT inner node is deciding between (e.g. *Animal vs. Vehicle*). (2) We find unseen classes that belong to the same superclass, from a different dataset. (e.g. Pull *Turtle* images from ImageNet). (3) Evaluate the model to ensure the unseen class is classified into the correct superclass (e.g. ensure *Turtle* is classified as *Animal*). For an NBDT, this is straightforward: one of the inner nodes classifies *Animal vs. Vehicle* (Sec 3.3). For a standard neural network, we consider the superclass that the final prediction belongs to. (*i.e.* When evaluating *Animal vs. Vehicle* on a *Turtle* image, the CIFAR-trained model may predict any CIFAR *Animal* class). See Appendix B.2 for details. Our NBDT consistently bests the original neural network by 8%+ (Table 6). When discerning *Carnivore vs. Ungulate*, NBDT outperforms the original neural network by 16%.

**Mid-Training Hierarchy**: We test NBDTs without using pre-trained weights, instead constructing hierarchies during training from the partially-trained network's weights. Tree supervision loss with mid-training hierarchies reliably improve the original neural network's accuracy, up to ∼0.6%, and the NBDT itself can match the original neural network's accuracy (Table 4). However, this underperforms NBDT (Table 1), showing fully-trained weights are still preferred for hierarchy construction.

## 5 INTERPRETABILITY

By breaking complex decisions into smaller intermediate decisions, decision trees provide insight into the decision process. However, when the intermediate decisions are themselves neural network

**Table 3: Comparisons of Losses.** Training the NBDT using tree supervision loss with a linearly increasing weight ("TreeSup(t)") is superior to training (a) with a constant-weight tree supervision loss ("TreeSup"), (b) with a hierarchical softmax ("HrchSmax") and (c) without extra loss terms. ("None"). $\Delta$ is the accuracy difference between our soft loss and hierarchical softmax.

| Dataset | Backbone | Original | TreeSup(t) | TreeSup | None | HrchSmax |
|---|---|---|---|---|---|---|
| CIFAR10 | ResNet18 | *94.97%* | **94.82%** | 94.76% | 94.38% | 93.97% |
| CIFAR100 | ResNet18 | *75.92%* | **77.09%** | 74.92% | 61.93% | 74.09% |
| TinyImageNet200 | ResNet18 | *64.13%* | **64.23%** | 62.74% | 45.51% | 61.12% |

**Table 4: Mid-Training Hierarchy.** Constructing and using hierarchies early and often in training yields the highest performing models. All experiments use ResNet18 backbones. Per Sec 3.4, $\beta_t, \omega_t$ are the loss term coefficients. Hierarchies are reconstructed every "Period" epochs, starting at "Start" and ending at "End".

| Hierarchy Updates | | | CIFAR10 | | | CIFAR100 | | |
|---|---|---|---|---|---|---|---|---|
| Start | End | Period | NBDT | NN+TSL | NN | NBDT | NN+TSL | NN |
| 67 | 120 | 10 | **94.88%** | **94.97%** | 94.97% | **76.04%** | **76.56%** | 75.92% |
| 90 | 140 | 10 | 94.29% | 94.84% | 94.97% | 75.44% | 76.29% | 75.92% |
| 90 | 140 | 20 | 94.52% | 94.89% | 94.97% | 75.08% | 76.11% | 75.92% |
| 120 | 121 | 10 | 94.52% | 94.92% | 94.97% | 74.97% | 75.88% | 75.92% |

predictions, extracting insight becomes more challenging. To address this, we adopt benchmarks and an interpretability definition offered by Poursabzi-Sangdeh et al. (2018): A model is interpretable if a human can validate its prediction, determining when the model has made a sizable mistake. To assess this, we adapt Poursabzi-Sangdeh et al. (2018)'s benchmarks to computer vision and show (a) humans can identify misclassifications with NBDT explanations more accurately than with saliency explanations (Sec 5.1), (b) a way to utilize NBDT's entropy to identify ambiguous labels (Sec. 5.4), and (c) that humans prefer to agree with NBDT predictions when given a challenging image classification task (Sec. 5.2 & 5.3). Note that these analyses depend on three model properties that NBDT preserves: (1) discrete, sequential decisions, so that one path is selected; (2) pure leaves, so that one path picks one class; and (3) non-ensembled predictions, so that path to prediction attribution is discrete. In all surveys, we use CIFAR10-trained models with ResNet18 backbones.

## 5.1 SURVEY: IDENTIFYING FAULTY MODEL PREDICTIONS

In this section we aim to answer a question posed in (Poursabzi-Sangdeh et al., 2018) *"How well can someone detect when the model has made a sizable mistake?"*. In this survey, each user is given 3 images, 2 of which are correctly classified and 1 is mis-classified. Users must predict which image was incorrectly classified given a) the model explanations and b) *without* the final prediction. For saliency maps, this is a near-impossible task as saliency usually highlights the main object in the image, regardless of wrong or right. However, hierarchical methods provide a sensible sequence of

**Table 5: Original Neural Network.** We compare the model's accuracy before and after the tree supervision loss, using ResNet18, WideResNet on CIFAR100, TinyImageNet. Our loss increases the original network accuracy consistently by $\sim .8 - 2.4\%$. NN-S is the network trained with the tree supervision loss.

| Dataset | Backbone | NN | NN-S |
|---|---|---|---|
| C100 | R18 | 75.92% | **76.96%** |
| T200 | R18 | 64.13% | **66.55%** |
| C100 | WRN28 | 82.09% | **82.87%** |
| T200 | WRN28 | 67.65% | **68.51%** |

**Table 6: Zero-Shot Superclass Generalization.** We evaluate a CIFAR10-trained NBDT (ResNet18 backbone) inner node's ability to generalize beyond seen classes. We label TinyImageNet with superclass labels (e.g. label *Dog* with *Animal*) and evaluate nodes distinguishing between said superclasses. We compare to the baseline ResNet18: check if the prediction is within the right superclass.

| $n_{class}$ | Superclasses | R18 | NBDT-S |
|---|---|---|---|
| 71 | Animal *vs.* Vehicle | 66.08% | **74.79%** |
| 36 | Placental *vs.* Vertebrate | 45.50% | **54.89%** |
| 19 | Carnivore *vs.* Ungulate | 51.37% | **67.78%** |
| 9 | Motor Vehicle *vs.* Craft | 69.33% | **77.78%** |

**Figure 4: CIFAR10 Blurry Images.** To make the classification task difficult for humans, the CIFAR10 images are downsampled by $4\times$. This forces at least partial reliance on model predictions, allowing us to evaluate which explanations are convincing enough to earn the user's agreement.

intermediate decisions that can be checked. This is reflected in the results: For each explainability technique, we collected **600** survey responses. When given saliency maps and class probabilities, only **87** predictions were correctly identified as wrong. In comparison, when given the NBDT series of predicted classes and child probabilities (e.g., "Animal (90%) $\rightarrow$ Mammal (95%)", without the final leaf prediction) **237** images were correctly identified as wrong. Thus, respondents can better recognize mistakes in NBDT explanations nearly 3 times better.

Although NBDT provides more information than saliency maps about misclassification, a majority – the remaining 363 NBDT predictions – were not correctly identified. To explain this, we note that $\sim 37\%$ of all NBDT errors occur at the final binary decision, between two leaves; since we provide all decisions except the final one, these leaf errors would be impossible to distinguish.

## 5.2 SURVEY: EXPLANATION-GUIDED IMAGE CLASSIFICATION

In this section we aim to answer a question posed in (Poursabzi-Sangdeh et al., 2018) *"To what extent do people follow a model's predictions when it is beneficial to do so?"*. In this first survey, each user is asked to classify a severely blurred image (Fig 4). This survey affirms the problem's difficulty, decimating human performance to not much more than guessing: **163** of **600** responses are correct (27.2% accuracy).

In the next survey, we offer the blurred image and two sets of predictions: (1) the original neural network's predicted class and its saliency map, and (2) the NBDT predicted class and the sequence of decisions that led up to it ("Animal, Mammal, Cat"). For all examples, the two models predict different classes. In 30% of the examples, NBDT is right and the original model is wrong. In another 30%, the opposite is true. In the last 40%, both models are wrong. As shown in Fig. 4, the image is extremely blurry, so the user must rely on the models to inform their prediction. When offered model predictions, in this survey, **255** of **600** responses are correct (42.5% accuracy), a 15.3 point improvement over no model guidance. We observe that humans trust NBDT-explained prediction more often than the saliency-explained predictions. Out of **600** responses, **312** responses agreed with the NBDT's prediction, **167** responses agreed with the base model's prediction, and **119** responses disagreed with both model's predictions. Note that a majority of user decisions ($\sim 80\%$) agreed with either model prediction, even though neither model prediction was correct in 40% of examples, showing our images were sufficiently blurred to force reliance on the models. Furthermore, 52% of responses agreed with NBDT (against saliency's 28%), even though only 30% of NBDT predictions were correct, showing improvement in model trust.

## 5.3 SURVEY: HUMAN-DIAGNOSED LEVEL OF TRUST

The explanation of an NBDT prediction is the visualization of the path traversed. We then compare these NBDT explanations to other explainability methods in human studies. Specifically, we ask participants to pick an expert to trust (Appendix, Figure 13), based on the expert's explanation – a saliency map (ResNet18, GradCAM), a decision tree (NBDT), or neither. We only use samples where ResNet18 and NBDT predictions agree. Of 374 respondents that picked one method over the other, **65.9%** prefer NBDT explanations; for misclassified samples, **73.5%** prefer NBDT. This supports the previous survey's results, showing humans trust NBDTs more than current saliency techniques when explicitly asked.

## 5.4 ANALYSIS: IDENTIFYING FAULTY DATASET LABELS

There are several types of ambiguous labels (Figure 5), any of which could hurt model performance for an image classification dataset like ImageNet. To find these images, we use entropy in NBDT

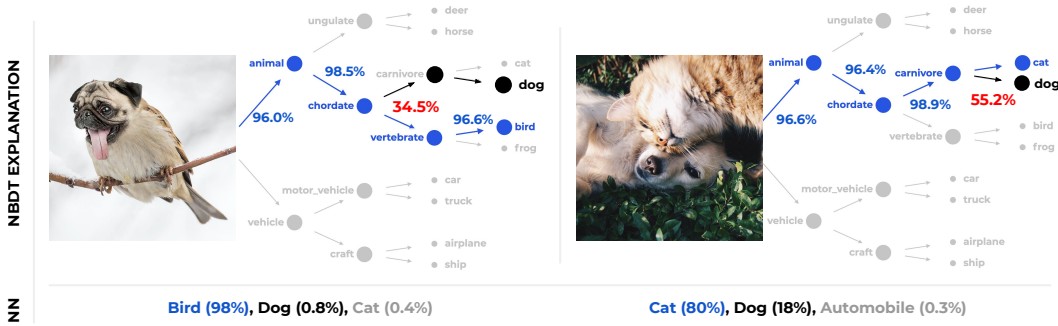

**Figure 5: Types of Ambiguous Labels.** All these examples have ambiguous labels. With NBDT (top), the decision rule deciding between equally-plausible classes has low certainty (red, 30-50%). All other decision rules have high certainty (blue, 96%+). The juxtaposition of high and low certainty decision rules makes ambiguous labels easy to distinguish. By contrast, ResNet18 (bottom) still picks one class with high probability. (Left) An extreme example of a "spug" that may plausibly belong to two classes. (Right) Image containing two animals of different classes. Photo ownership: "Spug" by Arne Fredriksen at gyyporama.com. Used with permission. Second image is CC-0 licensed at pexels.com.

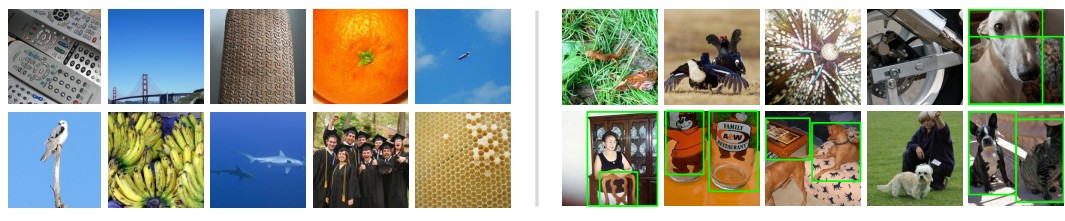

**Figure 6: ImageNet Ambiguous Labels.** These images suggest that NBDT path entropy uniquely identifies ambiguous labels in Imagenet, without object detection labels. We plot ImageNet validation samples that induce the most 2-class confusion, using TinyImagenet200-trained models. Note that ImageNet classes do not include people. (Left) Run ResNet18 and find samples that (a) maximize entropy between the top 2 classes and (b) minimize entropy across all classes, where the top 2 classes are averaged. Despite high model uncertainty, half the classes are from the training set – bee, orange, bridge, banana, remote control – and do not show visual ambiguity. (Right) For NBDT, compute entropy for each node's predicted distribution; take the difference between the largest and smallest values. Now, half of the images contain truly ambiguous content for a classifier; we draw green boxes around pairs of objects that could each plausibly be used for the image class.

decisions, which we find is a much stronger indicator of ambiguity than entropy in the original neural network prediction. The intuition is as follows: If all intermediate decisions have high certainty except for a few decisions, those decisions are deciding between multiple equally plausible cases. Using this intuition, we can identify ambiguous labels by finding samples with high "path entropy" – or highly disparate entropies for intermediate decisions on the NBDT prediction path.

Per Figure 6, the highest "path entropy" samples in ImageNet contain multiple objects, where each object could plausibly be used for the image class. In contrast, samples that induce the highest entropy in the baseline neural network do not suggest ambiguous labels. This suggests NBDT entropy is more informative compared to that of a standard neural network.

## 6 CONCLUSION

In this work, we propose Neural-Backed Decision Trees that see (1) improved accuracy: NBDTs out-generalize (16%+), improve (2%+), and match (0.15%) or outperform (1%+) state-of-the-art neural networks on CIFAR10, CIFAR100, TinyImageNet, and ImageNet. We also show (2) improved interpretability by drawing unique insights from our hierarchy, confirming that humans trust NBDT's over saliency and illustrate how path entropy can be used to identify ambiguous labels. This challenges the conventional supposition of a dichotomy between accuracy and interpretability, paving the way for jointly accurate *and* interpretable models in real-world deployments.

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
