# OpenReview forum: "NBDT: Neural-Backed Decision Tree"
_ICLR.cc/2021/Conference — ICLR 2021 Poster_

### Official Review · AnonReviewer1 · 2020-10-28
**Interesting results but unconvincing claims.**

**Rating:** 6
**Confidence:** 4

**Review:**

The paper proposes a method to make neural networks more accurate and interpretable by replacing their final layers with a probabilistic decision tree. As a result, the network can produce a sequence of decisions that leads to the final classification result, given an input image. The method is trained with soft decisions by assigning probabilities to each leaf, which are associated with a single class. The tree decision hyperplanes are constructed automatically from the backbone networks final dense layer and finetuned. The fact that decisions are soft solves the differentiablility problem of decisions as in various other similar papers, cited or uncited (more below).

The paper is not written very clearly, so it would be hard to reproduce. It's not clear in places if indices correspond to nodes or classes, as it is used interchangeably. The text misses a proper mathematical formulation of the operations done in inner nodes, and this all makes it difficult to understand what the loss is and how it can update the decisions in the tree. Perhaps it's possible to understand all the details by re-reading the text several times, but the paper definitely lacks clarity. A nice comparison would be with the Deep Neural Decision Forest (DNDF) paper. I'd expect that level of clarity from an ICLR paper.

The justification behind using the hyperplanes from the last layer of the CNN for constructing inner nodes is not explained. The visualization in Figure 2 indicates that the averages used for the parent nodes act like clusters - and somehow averaging them forms bigger clusters, but what really happens is that the rows of the final layer, which are unnormalized hyperplanes, are averaged to form new hyperplanes that are assumed to cover both classes, which would not be the case most of the time. I'm not sure if this has a reasonable geometric meaning, but the visualization gives the wrong idea.

The paper makes the claim: "Unlike previous decision trees or hierarchical classifiers, NBDTs use path probabilities for inference".
There are certainly many papers that use path probabilities for inference. In fact it's been the norm for discrete decisions, since hard decisions are much harder to formulate in a differentiable manner. DFDN uses path probabilities, as do older papers like "Decision Forests, Convolutional Networks and the Models in-Between" by Ioannou et al. (uncited). I don't understand this claim.
The biggest difference here is that each class gets exactly one dedicated leaf, instead of each leaf storing a distribution. It is not clear to me why this is a good idea though - it's clearly not preferred in the decision tree literature.

Another interesting point - most papers I know on the subject actually try to enable sparse activation at test time for efficiency, which is the harder problem to solve. See CondConv (Neurips 2019 - uncited), SplineNets (Neurips 2018 - uncited), Outrageously Large Neural Networks by Shazeer et al. (uncited), Conditional Information Gain by Bicici et al., etc. Activating all the branches of the tree or graph becomes prohibitively costly for deeper trees.

Interpretability is another major claim. The way it works is by using WordNet to assign higher level compound classes to images, such as Animal -> Turtle. This way by grouping similar leaves together, the inner nodes are assigned a meaning. This needs to be done when constructing the hierarchy, so the tree structure is manually given from what I understand. But then how are the pairs of nodes selected? It's not clear from the text, at least it is not explained clearly. In the Figure 5, a decision tree is given, which could only have been constructed by hand - i.e. with someone knowing that cats and dogs are the closest pairs. Where does this information come from for ImageNet when there are 1000 classes? WordNet?

Another concern about interpretability is that it is claimed that other good performing methods like DFDN are not interpretable. I don't see why that's the case if we use WordNet to assign meaning for the inner nodes of such solutions. I'd agree that it is not straightforward to do so, but with WordNet I can imagine how it could be done. So I'm not convinced about the claim that this solution is the most interpretable one when the paper does not explain why other solutions aren't in a persuasive manner.

One final concern is with the comparisons with similar models. The paper says these were reimplemented and tested with different backbones. Where are these implementations taken from, for instance for DFDN, which is quite complex? How can the reader trust these numbers?

I find the claims unconvincing and the results unpersuasive. I think the paper needs i) a better mathematical formulation to clarify the method, ii) better explanation of how the trees are constructed (e.g. if WordNet is used), iii) better understanding of differences from similar work. Currently I don't think it meets the bar for ICLR.


TreeSup result is likely wrong in Table 3.

After Rebuttal:

The authors did a great job in addressing most of the issues I have, and made many changes that helped with the clarity of the paper. There are still some remaining issues like Figure 2 assigning a wrong geometric meaning to the clusters formed by taking means of hyperplanes, and the uncited references, which are simply added to the references section (which should be fixed). But I think the added survey results are a great addition and a persuasive proof about the increased interpretabililty of these models. Therefore I'll increase my score to 6.

---

> ### Author Response · Authors · 2020-11-14
> **Interpretability, Related Work (Part 1)**
>
> **Interpretability and WordNet Assignment “It is claimed that other good performing methods like DNDF are not interpretable. I don't see why that's the case if we use WordNet to assign meaning for the inner nodes of such solutions”**
>
> It is not clear how WordNet meaning can be assigned to inner nodes for methods like DNDF, where leaves represent mixtures of classes. By contrast, each NBDT leaf represents a single class (e.g., Dog), making it possible to find the corresponding nodes in WordNet and compute the earliest ancestor of those two WordNet nodes (e.g., the ancestor of “Cat” and “Dog” is Animal, Sec 5.1).
>
> - **“In Figure 5, a decision tree is given, which could only have been constructed by hand… [How does it scale to] ImageNet? [Using] WordNet?”** Per Sec 3.2, we construct the hierarchy automatically, by running hierarchical agglomerative clustering in weight-space. The process is the same for ImageNet. To obtain meanings of each inner node, we assign WordNet meanings (Sec 5.1)
> - **“The visualization [of the induced hierarchy]... gives the wrong idea… not sure if this has a reasonable geometric meaning.”** Thank you for raising this point. We will need to either revisit this figure or the motivation in writing for the method. Preliminary experiments suggest similar results (downstream accuracy benefits), if we normalize the hyperplane normals before performing clustering.
>
> **Comparing with related work: “The paper needs… better understanding of differences from similar work… There are certainly many papers that use path probabilities for inference.”**
>
> In Sec 3.1, we state that our use of path probabilities comes from Deng et al (2012). We will also cite the papers mentioned from Kontscheider et al (2016). We should clarify that many (but not all) previous methods use hierarchical softmax (softmax per node), instead of a path probability from the root to the leaves (Sec 2 “Hierarchical Classification” last 2 sentences). In contrast to other works using path probabilities, our key differences were to not parameterize the inner nodes explicitly (meaning all classification architectures can be trained as NBDT without modifications), to keep leaves pure (to enable WordNet assignment, Sec 2), and to employ a hierarchy built from neural network weights rather than input data (for improved accuracy, Sec 3.2).
>
> - **“Where are these implementations [of related work] taken from, for instance for DNDF, which is quite complex? How can the reader trust these numbers?”** Numbers (Table 1) were taken from the original papers, except for 3 methods that did not report top-1 accuracy on comparable datasets: classic DTs, oblique DTs, and DNDF. DNDF authors did not provide top-1 accuracies on CIFAR or ImageNet (top-5 reported). However, given the popularity of the work, we had to compare with DNDF results somehow: 1) The DNDF authors did not open-source code. 2) Thus, we used a third-party implementation of DNDF that we are not affiliated with https://github.com/jingxil/Neural-Decision-Forests. The original results with LeNet were low, so we replaced the backbone with ResNet18 to improve results by 5%+. The two DTs have commonly-accepted implementations in scikit-learn, OC1 and were run on outputs of pre-trained backbones from torchvision.
> - **Most papers I know on the subject actually try to enable sparse activation at test time... Activating all the branches of the tree or graph becomes prohibitively costly for deeper trees.** We show results for a conditionally-executed variant of NBDT (NBDT-H) in Appendix Table 6. Although this variant also outperforms alternative decision-tree-and-neural-network hybrids, we relegated it to the appendix for a few reasons. We can cite the mentioned papers as suggested, but we believe this is out of the scope of our current work:
>     - We agree that test-time computational efficiency is also an important and challenging task. Not only that but enabling k-way classification for extremely large k. However, we chose to narrow the scope and focus on accuracy and interpretability on more commonplace classification datasets.
>     - Unlike these conditionally-executed models, as we show in Fig 1 C, the NBDT can recover from highly-uncertain, incorrect predictions (Sec 3.1 last paragraph). This accounts for the NBDT’s 2-3% win over its conditionally-executed variant.

---

> ### Author Response · Authors · 2020-11-14
> **Clarity (Part 2)**
>
> **Clarity: “The paper is not written very clearly, so it would be hard to reproduce. A nice comparison would be with the Deep Neural Decision Forest (DNDF) paper. I'd expect that level of clarity from an ICLR paper.”**
>
> While the first and second reviewers (from the topmost to bottom-most reviews) found the paper clear, we will open-source all code needed to reproduce our results. We are also happy to clarify any confusions in the paper, starting with the suggestions made in these reviews. We currently already compare against DNDF in Table 1 and show that NBDT outperforms DNDF by a large margin across datasets.
>
> - **“The text misses a proper mathematical formulation of the ... inner nodes”** We have a formulation in Sec 3.1 but recognize the definition of r_i is unclear. We will clarify Sec 3.1.
> - **“It's not clear in places if indices correspond to nodes or classes”** We will make this explicit and amend Sec 3. Node indices are denoted using i (Eq 2) and class indices are denoted using lowercase c (Eq 3).
> - **“TreeSup results is probably wrong in Table 3”** Thank you for catching this. TreeSup in Table 3 for CIFAR10 should be 94.76%, making TreeSup (t) the best result.

---

> ### Author Response · Authors · 2020-11-25
> **Manuscript Update**
>
> Thank you for your feedback and suggestions. We rewrote major portions of Sec 3 Methods, incorporating your suggestions regarding technical correctness and clarity. We have also added the recommended citations.

---

### Official Review · AnonReviewer3 · 2020-10-28
**The paper presents an algorithm (NBDT) for interpretable image classification, demonstrating a higher preference rate (65.9%) when compared to a saliency map for the model explanation among 374 participants.**

**Rating:** 6
**Confidence:** 5

**Review:**

Significance:
This article seems to be a useful contribution to the literature on interpretable deep networks. However, the paper could be strengthened by demonstrating and analyzing the interpretability of approaches to other types of data, such as sequential data.

Novelty:
This paper's main contribution is to offer a new hybrid model that combines a deep neural network with a tree. The authors used the weights of the last layer of a DNN to build a tree from bottom to top, where each inner node in the tree is the average weight of each child. Then, the authors used softmax to compute the probability of  routing for each child.

Potential Impact:
The approach presented in this paper is well-evaluated in computer vision but potentially useful in many other settings.

Technical Quality:
+ The technical content of the paper appears to be correct, albeit there is some room for improvement.
+ Page 2, the authors said, "These models likewise limit interpretability by supporting no more than depth-2 trees." Having the depth-2 tree actually improves the interpretability since it is easier to follow the model prediction. For example, a tree of depth-2 considers more interpretable than a tree of depth-4. The authors should rephrase this sentence.
+ While the authors claim that linearly increasing the weight in NBDT is the superior method, why the NBDT with the constant rate overperforms on CIFAR-10 as shown in Table 3?
+ Based on the NBDT's explanation, all leaf nodes should have the same depth, but in the example shown in Figure 6 and supplement, the leaf nodes are in different depths. The authors need to explain why the final tree has leaf nodes with different depths.
+ The authors did not compare their method's interpretability with a similar methods, such as Adaptive Neural Trees (Tanno et al. 2019).  I suggest running this experiment since Adaptive Neural Trees has a different interface than NBDT. . While the number of participants is noticeable in the interpretability study, it seems that participants only answer one question. Adding more questions could strengthen the paper. Further, the authors should provide a summary of participants (e.g., age, education, and gender).
+ The paper will be strengthened if the authors run an experiment without using a pre-trained neural network on a small dataset like MNIST to demonstrate their algorithm's effectiveness.

Presentation/Clarity
+ While the paper is fairly readable, there is room for improvement in the clarity.
+ Page 3, the last paragraph forgot a period after the parentheses. "path (Figure 1 C, Appendix Table 6) This ..."
+ Page 7, figure 4, I believe that the authors mean "without NBDT" instead of "without ResNet."
+ While the authors explained how they label the tree nodes with WordNet in the supplement, there is no explanation in the main paper. Labeling the tree's nodes is an important part of the algorithm and should be included in the main paper.
+ Page 5, the authors used "Hierarchy Ablation" and "Loss Ablation" subtitle in the paper. The word "Ablation" seems inappropriate in this context.

Reproducibility
The paper describes all the algorithms in full detail and provides enough information for an expert reader to reproduce its results. However, the authors did not discuss when they start to increase the W in equation 3, how to determine to stop the W from increasing, and with what rate the W should be increased.

---

> ### Author Response · Authors · 2020-11-14
> **Expanded Survey, Experiments without Pre-trained Weights**
>
> **“Adding more [survey] questions could strengthen the paper.” Authors should compare with similar methods like ANT and provide a summary of participant demographics].**
>
> We will run further human studies in light of both suggestions, to a) compare with ANT and b) ask more questions to test different aspects of interpretability. For our surveys, we used Amazon’s Mechanical Turks, which doesn’t provide demographic information by default, but we will explore alternatives.
>
> **The paper will be strengthened if the authors run an experiment without using a pre-trained neural network on a small dataset like MNIST to demonstrate their algorithm's effectiveness.**
>
> This is an interesting idea; thank you for the recommendation. We are now running experiments without a pre-trained neural network to test the idea (hierarchy is built from the currently-trained model’s weights, partway through training). We will follow-up with method details and results on MNIST and CIFAR as they become available.
>
> **Technical Clarity and Presentation**
>
> - **A tree of depth-2 is more interpretable than a tree of depth-4.** Thank you for pointing this out -- we agree. We should clarify to instead say that for many classes, depth-2 trees are suboptimal, because each decision rule will then have too many options to pick from. E.g., A 512-way decision is difficult to interpret. In the limit, we could consider a fully-connected layer a depth-1 tree, and our hope is to distance ourselves from decision rules with too many options.
> - **All leaf nodes should have the same depth, but in the example shown in Figure 6 and supplement, the leaf nodes are in different depths.** Since we use hierarchical agglomerative clustering (Sec 3.2), our trees can have leaves at different levels. For example, say classes 1, 2, and 3 are extremely close but class 4 is far. Classes 1 and 2 could be clustered first. Then, 1-2 is clustered with 3. Finally, 1-2-3 is clustered with 4. This creates an imbalanced binary tree. We will clarify this in the manuscript, as suggested.
> - **Various Presentation Notes: Forgot a period, mean “without NBDT” instead of “without ResNet”, word “Ablation” seems inappropriate** Thank you for noting these. We will amend the manuscript to incorporate all your suggestions.
> - **Labeling the tree's nodes ... should be included in the main paper.** We briefly mention the node labeling strategy in Sec 5.1 but will move details from the supplement to the main manuscript as suggested.
> - **While the authors claim that linearly increasing the weight in NBDT is the superior method, why the NBDT with the constant rate overperforms on CIFAR-10 as shown in Table 3?** Thank you for catching this. TreeSup in Table 3 for CIFAR10 should be 94.76%, making TreeSup (t) the best result.

---

> > ### Author Response · Authors · 2020-11-24
> > **Results: Additional Surveys, Results without Pretrained Weights**
> >
> > Thank you for your patience and update; per your and Reviewer 2’s request, we have run 2 more human experiments to solidify our claim of interpretability. Since we are focusing specifically on how much humans trust our NBDTs decisions, we refer to Reviewer 2’s comment that human trust can be better measured by having humans complete tasks with the model. We are currently working on manuscript updates; in the interim, here are our results:
> >
> > **More Human Studies**
> >
> > We conducted 2 more human studies aimed at answering the following questions raised in [3]: (1) “How well can people detect when a model has made a sizable mistake?” and (2) ”To what extent do people follow a model’s predictions when it is beneficial to do so?”.
> >
> > **Study 1: “How well can people detect when a model has made a sizable mistake?”**
> >
> > To address the first question, each user is given 3 images, 2 of which are correctly classified and 1 is mis-classified. We use ResNet18 trained and evaluated on CIFAR10. Users must predict which image was incorrectly classified given a) the model explanations and b) *without* the final prediction. For saliency maps, this is a near impossible task as saliency usually highlights the main object in the image, regardless of wrong or right. However, hierarchical methods provide a sensible sequence of intermediate decisions that can be easily checked to see if they match the expected result. This is reflected in the results: For each explainability technique, we collected 600 survey responses. When given saliency maps and class probabilities, only 87 predictions were correctly identified as wrong. In comparison, when given the NBDT series of predicted classes and child probabilities (e.g., “Animal (90%) -> Mammal (95%)”, without the final leaf prediction) 237 images were correctly identified as wrong.
> >
> > One note: Although NBDT provides more information than saliency maps about misclassification, a majority -- 363 predictions -- were not correctly identified. To explain this, we note that ~37% of all NBDT errors occur at the final binary decision, between two leaves; since we provide all decisions except the final one, these leaf errors would be impossible to distinguish.
> >
> > **Study 2: ”To what extent do people follow a model’s predictions when it is beneficial to do so?”**
> >
> > To address the second question, each user is given a blurred image (kernel size 5x5) and two sets of predictions: (1) the original neural network’s predicted class and its saliency map, and (2) the NBDT predicted class and the sequence of decisions that led up to it (“Animal, Mammal, Cat”). For all examples, the two models predict different classes. In 30% of the examples, NBDT is right and the original model is wrong. In another 30%, the opposite is true. In the last 40%, both models are wrong. Since the image is extremely blurry (http://people.eecs.berkeley.edu/~alvinwan/nbdt/cifar10-downsampled/0-img-downsampled.jpg), the user must rely on the models to inform their prediction. From this survey we show that humans choose to rely on the NBDT-explained prediction more often than the saliency-explained predictions. Out of 600 responses, 312 responses chose the NBDT’s prediction, 167 responses chose the base model’s prediction, and 119 responses chose neither model’s predictions. Note that a majority of user decisions (~80%) agreed with either model prediction, even though neither model prediction was correct in 40% of examples, showing our images were sufficiently blurred to force reliance on the models. When both models are incorrect, ~45% of responses agree with NBDT, ~34% with the original model, and 22% with neither.
> >
> > **Experiments using a partially trained network**
> >
> > If the hierarchy is induced part way through training and then trained for the rest of the time with the TreeSupLoss, accuracy for the NBDT using soft inference is ~0.15% better than the original network, while the original network’s accuracy is boosted by ~0.6%. These results were obtained using ResNet18 on CIFAR10. This shows that removing the requirement of pretrained weights has promise but that a hierarchy built at the end of training is preferred, for supervision during training.
> >
> > *We note that in the original review, you had requested a comparison with Adaptive Neural Trees. We were unable to reproduce Adaptive Neural Trees in time due to technical limitations (out-of-date PyTorch with incompatible CUDA) but will continue working on this accordingly, for the appropriate hierarchy comparison you recommended*
> >
> > [3] Poursabzi-Sangdeh, et al. Manipulating and Measuring Model Interpretability. https://arxiv.org/abs/1802.07810.

---

> > ### Author Response · Authors · 2020-11-25
> > **Manuscript Update**
> >
> > Thank you again for your feedback and suggestions. We have incorporated the surveys in a revised Sec 5 (Interpretability), mid-training hierarchy into Sec 4 (Analysis), and the other suggestions for technical and presentation clarity. We have also rewritten major portions of Sec 3 (Methods) to hopefully address the concerns regarding clarity and technical correctness.

---

### Official Review · AnonReviewer2 · 2020-10-29
**Compelling interpretability methodological work but minor flaws in motivation, lack of discussion about practical limitations**

**Rating:** 7
**Confidence:** 4

**Review:**

The authors did a fantastic job of answering questions, revising their manuscript in accord with reviewer feedback (Sec 3.4 title), and even adding new experimental results based on reviewer suggestions (mid-training hierarchy) and reflecting best practices in interpretability research. I was really impressed by their nimbleness and responsiveness. I will raise my score to a 7: I think this is a very solid paper and excellent research effort around a nascent idea. In particular, I think its impact is limited by

- its close coupling to naturally hierarchical problems, e.g., multi-class classification with a taxonomy
- its close coupling to image data and tasks
- its heuristic nature: fully train neural net, infer hierarchy via clustering, retrain neural net, then map a priori labels onto inferred hierarchy

The "10" version of this paper (maybe future work?) would propose a way to infer the hierarchy on the fly and show how to apply it onto other kinds of data and problems with different structures.

-----

This submission proposes a modification of neural networks that replaces the "final linear layer with a decision tree." The term "decision tree" is applied somewhat loosely to a hierarchical neural architecture akin to a hierarchical softmax. In the current work (as I understand it), this hierarchy is induced from a pre-trained multi-output, e.g., multiclass, neural network via a hierarchical clustering and subsequent averaging of the output weights. At inference time, path probabilties can be computed based on the chain rule. Predictions can be made based on either a greedy traversal of the tree (choosing the most likely child at each step, a la hierarchical softmax) or by choosing the most probable leaf, which requires computing all path probabilities. Empirical results across three standard image datasets are suggestive, if not conclusive, and the paper concludes with some interesting, albeit cursory, examples of potential "interpretability" applications.

The submission summarizes its contributions at the end of Section 1 as follows:
1. It proposes a tree-structured loss to augment supervised neural network training (predominantly for multiclass classification problems).
2. It describes a heuristic to induce a hierarchy in the output weights of a pre-trained multi-output neural network, enabling decision tree-like inference and provide evidence it is more effective than other approaches for inducing hierarchies.
3. It presents simple case studies of how the induced hierarchy can be used for traditional "interpretability" tasks, like debugging and generating explanations.

I appreciate the idea at the center of this paper -- adding simple hierarchical structure to a multi-output neural network, with the aim of increased interpretability -- but I feel the work as it is presented is nascent and the manuscript itself is flawed. I lean toward rejection at the moment, but I could be persuaded to change my mind by some combination of solid revisions, convincing author response, or vociferous advocacy from other reviewers.

I will briefly extol the paper's strengths before providing a longer discussion of what I consider to be its key weakness. First, I really like the last sentence in the paper:

"This challenges the conventional supposition of a dichotomy between accuracy and interpretability, paving the way for jointly accurate and interpretable models in real-world deployments."

Weaknesses in the evaluation of its interpretability claims aside, I agree with this statement. I think the case studies presented do provide evidence of improved interpretability alongside small accuracy improvements. I think this paper does succeed in demonstrating that accuracy and interpretability are not necessarily competing objectives, at least for certain tasks (multiclass classification of images).

A laundry list of other strengths:

- The motivation is strong (modulo weakness discussed below): there is a growing need to provide human-understandable insights into decisions made by complex machine learning models.
- The proposed approach is simple and elegant, easy to implement, and empirically effective. I'm quite impressed that the proposed tree loss appears to improve accuracy (!) on multiple tasks.
- I also think this paper lays groundwork for a direction of research that the community could continue to build on.

I think that the manuscript's largest flaw, ironically, regards interpretability, its primary motivation. The work's central claim is that the tree-structured decision layer delivers improved interpretability with comparable or slightly improved accuracy. In its discussion of this claim, the manuscript provides no precise definition of "interpretable," making it difficult to verify the claim qualitatively or quantitatively. Section 5 presents a vignette of case studies, but the discussion of each is quite limited. In particular, none of the use cases is fully motivated or placed in the context of previous research on interpretability definitions [1][2]. The cursory presentation of results for each do the results a disservice by making it difficult for the reader to recognize and assess their significance.

To quote the introduction from Lipton's _The Mythos of Model Interpretability_ [1],

"Despite the absence of a definition, papers frequently make claims about the interpretability of various models. From this, we might conclude that either: (i) the definition of interpretability is universally agreed upon, but no one has managed to set it in writing, or (ii) the term interpretability is ill-defined, and thus claims regarding interpretability of various models may exhibit a quasi-scientific character."

I believe the paper would be strengthened by focusing on one use case, e.g., debugging or human trust, using the ~1 page dedicated to Section 5 to motivate it more fully and to present the results in detail. If the primary use case is generalization or debugging, then I suggest designing a quantitative analysis so defend against claims of cherry picking the best results (a common problem in presenting "example" interpretability results).

Section 5.4 includes a quantitative evaluation, but I question whether mere human preference is evidence of "human trust." More recent research on trust appear to use more elaborate studies in which trust is measured by subjects' rate of success in performing a particular task aided by the machine learning model [3].

I want to caveat the above: I really appreciate this line of work and think it has value. There is an ongoing discussion in our community about rewarding good ideas, rather than punishing imperfect or incomplete execution. I also acknowledge that I am far from an expert in the latest interpretability research. Nonetheless, my understanding is that interpretability researchers have grown more skeptical of interpretability claims about new methods absent a rigorous framework (definitional and/or experimental) for evaluating those claims.

When I read this paper, I find it hard to escape the conclusion that its interpretability claims rest on the presupposition that trees are naturally more interpretable (and further that readers will accept this dogma). I disagree with this assertion (see below), but even if it were generally true, I still think the paper would be strengthened by adding a more rigorous discussion and analysis of its claims. Propose a definition or criterion (see [1][2] for ideas), ideally one that could be assessed qualitatively and evaluated empirically, then apply it.

Regarding the claim about trees in Section 5: "The interpretability of a decision tree is well-established when input features are easily understood (e.g. tabular data in medicine or finance)."

I would dispute that this is "well-established" for anything but the simplest decision tree models, with a single tree consisting of a small number of splits using a handful of features, which are rare in realistic settings. The most commonly used tree-structured models (gradient boosted decision trees and random forests) are not readily interpretable, even for tabular data and especially for high dimensional inputs. This has made research like SHAP [4] of great interest to practitioners.

What is more, even for tabular data, the neural decision trees described in this paper are (to my understanding) basically a cascade of linear classifiers, with split each having access to all features at once. This does do not lend itself to the same kind of "interpretation" one gets for classic decision trees that use one feature per split. With even modestly deep hierarchies, the resulting "explanations" would rapidly become quite complex.

I see one other weakness in the proposed method itself: as I understand things, it requires access to a pretrained neural network. At the very least, one needs pre-existing output weights to cluster in order to induce a hierarchy -- and the induced hierarchy is a necessary component in the presented results. This isn't a fatal flaw -- learning a hierarchy on the fly could be left for future work. Nonetheless, it limits the work's usefulness and potential impact.

What is more, I don't think the manuscript is sufficiently clear about this requirement: on my first pass through the paper, I came away with the impression that there was a way to learn the hierarchy while training the neural net -- the inclusion of a section entitled "Training with Tree Supervision Loss" seems to imply this. I suggest revising the text to make it crystal clear that it is not possible to use the tree loss to train a neural net from scratch -- at least, not without a predefined hierachy (perhaps from a previous training run or prior knowledge).

I will now summarize the improvements I suggest for strengthening the manuscript:
1. Focus on one definition of interpretability and then analyze central claims through that lens. Introduce it early in the paper (introduction) and then dedicate Section 5 to it, rather than trying to cover lots of use cases superficially.
2. Make the limitations of the proposed approach VERY clear. In particular, you need a predefined hierarchy to train with the tree loss, and you need pretrained neural net (or pre-existing weights, at least) to induce a hierarchy based on clustering.
3. If the intention is for this approach to be used exclusively for finetuning or adapting an existing neural network, then this should be made clear in the text. Consider renaming Section 3.3.
4. Justify (or reword) statements like "the interpretability of a decision tree is well-established" or "neural features are visually interpretable" (a single reference does not suffice...the Olah distill survey draws no such definitive conclusions).

I have a few questions:
- One thing that is not clear: when training with tree loss, are weights shared across nodes? In particular, the weight vector for an inner node is the average of its descendent leaf node weight vectors. When training with tree loss, do we then treat that inner node weight vector as a set of independent parameters with separate updates? Or do we continue to treat it as a sum of leaf parameters, so that leaf and inner node updates affect the same parameters, as in an RNN or recursive network.
- Is there possibly a heuristic that could approximate "learning the hierarchy?" For example, train with a basic loss for enough iterations until the output weights start to converge, then pause training to induce the hierarchy and then resume training with the tree loss. Part of the heuristic could be guidance about how to detect when the output weights have sufficiently converged.
- What are the key differences between this approach and a hierarchical softmax? My understanding is that they're basically equivalent at inference time (except maybe traditional hierarchical softmax uses hard decisions?). What about during training? Is it maybe the use of negative sampling for hierarchical softmax?
- How would this approach perform for extremely high dimensional output spaces -- one of the primary motivations for hierarchical softmax? I imagine that for some output cardinality, "soft" inference becomes computationally infeasible.

[1] Lipton. The Mythos of Model Interpretability. https://arxiv.org/abs/1606.03490.
[2] Doshi-Velez and Kim. Towards A Rigorous Science of Interpretable Machine Learning. https://arxiv.org/abs/1702.08608.
[3] Poursabzi-Sangdeh, et al. Manipulating and Measuring Model Interpretability. https://arxiv.org/abs/1802.07810.
[4] Lundberg, et al. From local explanations to global understanding with explainable AI for trees. https://www.nature.com/articles/s42256-019-0138-9.

---

> ### Author Response · Authors · 2020-11-14
> **Interpretability Definition, Making Limitations Explicit, Rewording**
>
> **Suggestion 1. Focus on one definition of interpretability and then analyze central claims through that lens.**
>
> Thank you for the suggestion; we are actively following this: in the context of previous interpretability definitions, we will pick an aspect of interpretability to focus on, then evaluate it qualitatively and quantitatively (likely by redoing the surveys to be task-oriented, as suggested). We will follow up with a longer response shortly.
>
> **Suggestion 2 + 3. Make the limitations of the proposed approach VERY clear. (You need pre-existing weights). If exclusively made for fine-tuning, consider renaming Section 3.3.**
>
> Thank you for pointing this out. We understand now that this is not explicit enough and will amend descriptions in the introduction, Sec 3 (intro paragraph), and Sec 3.2 to make this clearer -- a sentence to the tune of “This induced hierarchy requires a pre-trained neural network’s weights”.
>
> Per your suggestion below, we will also run experiments without a pre-trained network, generating the hierarchy with weights from the model currently being trained, partway through training. If these experiments are successful, we will make clear there are two training strategies: (1) fine-tuning, which requires a pre-trained network for the hierarchy (and possibly weight initialization) and (2) training from scratch, without using a pre-trained network for the hierarchy or weight initialization.
>
> **Suggestion 4. Justify (or reword) statements like "the interpretability of a decision tree is well-established".**
>
> These are fair critiques, and we will reword this section. We agree that the simplest of decision trees with tabular data is the only scenario where a decision tree is directly interpretable. Our hope was to illustrate why it is not straightforward to show NBDT interpretability, due to higher-dimensional inputs, motivating our analysis in Sec 5.
>
> **Questions**
>
> - **When training with tree loss, are weights shared across nodes?** Indeed, any two inner nodes, where one is the ancestor of the other, will share weights. We will clarify this in Sec 3.1.
> - **Is there a heuristic that could approximate "learning the hierarchy?” (+idea)** This is an interesting idea; thank you for the suggestion. We are now running experiments to test hierarchies built partway through training. We will follow-up with results as they become available.
> - **What are the key differences between this approach and a hierarchical softmax?** At test time, hierarchical softmax will indeed make a hard decision at each node. (This means any incorrect inner decision leads to an incorrect prediction. By contrast, the “soft” inference could tolerate incorrect but highly-uncertain inner decisions, Fig 1 C) During training, hierarchical softmax (HS) uses one softmax term for each node (Appendix C). We compare HS training with our “soft” tree loss in Appendix Table 7, where HS achieves 0.5-2.5% lower accuracy than tree loss. (Apologies, the columns are shifted forward by 1. “None” should be 94.32%, “TSL” should be 94.50% and “HS” should be 93.94%) Conceptually, we believe that the tree loss attains better accuracy than hierarchical softmax, because our loss disproportionally up-weights the decisions higher up in the tree, thus putting a greater emphasis on learning high level concepts (Sec 3.3).
> - **How would this approach perform for extremely high dimensional output spaces -- one of the primary motivations for hierarchical softmax? I imagine that for some output cardinality, "soft" inference becomes computationally infeasible.** We concur that “soft” inference would not scale (e.g., to a million-way classification problem). Soft inference scales only as far as a regular classification neural network could. There are two possibilities we have in mind:
>     1. One possibility is to use just the NBDT’s induced hierarchy (but with hierarchical softmax and “hard” inference). We describe the process in Appendix C and show results in Table 7. Relative to the original neural network, accuracy degrades, but Table 2 suggests the induced hierarchy at least obtains higher-accuracy than other hierarchies
>     2. Another possibility we did not explore is executing a “locally-soft” inference and training. One possible manifestation is to cluster several inner nodes in the tree into a “super node”. Apply HS and “hard” inference between “super nodes” but within each supernode, use soft inference and tree loss. The limits of a neural network (probably around several thousand classes) could determine the “super node” size.

---

> > ### Comment · AnonReviewer2 · 2020-11-14
> > **Thanks for the thoughtful response!**
> >
> > Hey authors, just wanted to let you know that I read and appreciated your thoughtful response. I especially want to commend you on receiving the constructive feedback with an open mind and humility -- while nonetheless advocating for your work, as you should. I look forward to hearing about the results from the "learning the hierarchy" heuristic and to reading the updated manuscript. In the meantime, I plan to re-read the submission with your responses in mind and will post any follow up questions that come to mind.

---

> > ### Comment · AnonReviewer2 · 2020-11-23
> > **Updated manuscript or results?**
> >
> > Hey authors, just wanted to remind you that the author discussion period ends tomorrow (November 24...not sure what time). Don't forget to upload your updated manuscript and to share the results from the mid-training hierarchy construction! If you find yourself in a time crunch, I think the priority for me would be any new results you have.

---

> > > ### Author Response · Authors · 2020-11-24
> > > **Results (Part 1): Mid-Training Hierarchy + Surveys**
> > >
> > > Thanks for the reminder! Per your suggestions, we have (1) picked a definition of interpretability using previous work, (2) run user studies to support our claim to interpretability, under this definition, and (3) run the mid-training hierarchy experiments. We are currently working on manuscript updates; in the interim, here are our results:
> > >
> > > **1. Mid-training hierarchy improves accuracy but sees less improvement than pre-trained hierarchy**
> > >
> > > If the hierarchy is induced part way through training and then trained for the rest of the time with the TreeSupLoss, accuracy for the NBDT using soft inference is ~0.15% better than the original network, while the original network’s accuracy is boosted by ~0.6%. These results were obtained using ResNet18 on CIFAR10.
> > > - If we construct the hierarchy only once midway through training, accuracy decreases by ~1%, so we instead reconstruct the hierarchy every 10 epochs for our final result.
> > > - As you suggested, tree supervision is best performed after hierarchy has stabilized. Accuracy can change by up to 1%, by changing which epoch the hierarchy is first constructed at.
> > >
> > > From this we can conclude that pre-trained weights for the hierarchy are preferred over weights obtained mid-training.
> > >
> > > **2. A model is interpretable if a user can tell if a model has made an incorrect prediction.**
> > >
> > > After reading the suggested related works, we define a model to be interpretable if it is able to indicate to the user that either it has made a mistake or the input is significantly different from the training data. This increases humans’ trust in the model, which can be measured by how well they perform on certain tasks with the model's assistance, as you mentioned. Thus, we show that NBDT’s are more interpretable than other methods via human studies as well as show an application of this method for dataset debugging.
> > >
> > > In our additional human studies, we aim to answer the following questions raised in [3] that are directly applicable to computer vision: (1) “How well can people detect when a model has made a sizable mistake?” and (2) ”To what extent do people follow a model’s predictions when it is beneficial to do so?”. In all surveys, the images are viewed at a larger size, 150x150, even though the image itself is low-resolution (from CIFAR10).
> > >
> > > **3. Study: “How well can people detect when a model has made a sizable mistake?”**
> > >
> > > To address the first question, each user is given 3 images, 2 of which are correctly classified and 1 is mis-classified. We use ResNet18 trained and evaluated on CIFAR10. Users must predict which image was incorrectly classified given a) the model explanations and b) *without* the final prediction. For saliency maps, this is a near impossible task as saliency usually highlights the main object in the image, regardless of wrong or right. However, hierarchical methods provide a sensible sequence of intermediate decisions that can be easily checked to see if they match the expected result. This is reflected in the results: For each explainability technique, we collected 600 survey responses. When given saliency maps and class probabilities, only 87 predictions were correctly identified as wrong. In comparison, when given the NBDT series of predicted classes and child probabilities (e.g., “Animal (90%) -> Mammal (95%)”, without the final leaf prediction) 237 images were correctly identified as wrong.
> > >
> > > One note: Although NBDT provides more information than saliency maps about misclassification, a majority -- 363 predictions -- were not correctly identified. To explain this, we note that ~37% of all NBDT errors occur at the final binary decision, between two leaves; since we provide all decisions except the final one, these leaf errors would be impossible to distinguish.

---

> > > ### Author Response · Authors · 2020-11-24
> > > **Results (Part 2)**
> > >
> > > (continued from previous comment)
> > >
> > > **4. Study: “To what extent do people follow a model’s predictions when it is beneficial to do so?”**
> > >
> > > To address the second question, each user is given a blurred image (kernel size 5x5) and two sets of predictions: (1) the original neural network’s predicted class and its saliency map, and (2) the NBDT predicted class and the sequence of decisions that led up to it (“Animal, Mammal, Cat”). For all examples, the two models predict different classes. In 30% of the examples, NBDT is right and the original model is wrong. In another 30%, the opposite is true. In the last 40%, both models are wrong. Since the image is extremely blurry (http://people.eecs.berkeley.edu/~alvinwan/nbdt/cifar10-downsampled/0-img-downsampled.jpg), the user must rely on the models to inform their prediction. From this survey we show that humans choose to rely on the NBDT-explained prediction more often than the saliency-explained predictions. Out of 600 responses, 312 responses chose the NBDT’s prediction, 167 responses chose the base model’s prediction, and 119 responses chose neither model’s predictions. Note that a majority of user decisions (~80%) agreed with either model prediction, even though neither model prediction was correct in 40% of examples, showing our images were sufficiently blurred to force reliance on the models. When both models are incorrect, ~45% of responses agree with NBDT, ~34% with the original model, and 22% with neither.

---

> > > ### Author Response · Authors · 2020-11-25
> > > **Manuscript Update**
> > >
> > > Thank you again for your feedback and suggestions. We have incorporated the surveys in a revised Sec 5 (Interpretability), mid-training hierarchy into Sec 4 (Analysis), and the other suggested revisions -- including a more specific interpretability definition in the introduction.

---

### Official Review · AnonReviewer4 · 2020-10-29
**A Nice Paper**

**Rating:** 6
**Confidence:** 2

**Review:**

This paper proposes a neural-backed decision tree that aims to improve both the accuracy and the interpretability of deep learning models. Training under a newly introduced tree supervision loss, the authors show that NBDTs can outperform and out-generalize some modern architectures on several image datasets.

Overall this paper is well written and established. The idea of using a differentiable oblique decision tree to replace the final linear layer is interesting. The authors provide clear illustration of the procedure and promising experimental results.

Questions:

1. What is the main intuition that NBDTs can outperform the original network?
2. Given the classes are in the leaves, does the ordering of classes in the leaf layer matter? How should one determine which two classes are in the same bottom subtrees?

Minor comments:

1. Figure 2 Step A: y_d -> y_k or y_4.
2. Why are there many n/a results in Table 1?
3. Section 3.1 in the Compute node probabilities paragraph, the definition of r_i seems confusing.
4. What does NN mean in Table 1? How different is it from CNN-RNN?

---

> ### Author Response · Authors · 2020-11-14
> **Intuition for Improvement, Leaf Ordering**
>
> **“What is the main intuition why NBDTs outperform the original network?”**
>
> We believe NBDTs outperform the original neural networks by learning higher-order concepts, explicitly (like Animal, Vehicle). This is supported by the generalization results in Table 5, where NBDTs see disproportionate accuracy boost higher up in the hierarchy (9-16%); this is in stark contrast to the overall accuracy boost of 1-2%, suggesting higher-order decisions dominate the improvements.
>
> **“Does the ordering of the classes in the leaf layer matter?” How do you determine which pairs of classes are grouped together?**
>
> As you suggest, the ordering of the leaf layer is unimportant. The two classes with the closest weight embeddings are grouped together. The next two closest are also grouped together. And so on and so forth. In short, we run hierarchical agglomerative clustering (Sec 3.2), so that iteratively, the closest two clusters are combined.
>
> **Clarifications**
> - **Figure 2 Step A: $y_d$ -> $y_k$ or $y_4$.** Thank you for the suggestion; we will change the subscript to a k or 4, as suggested.
> - **Why are there so many N/A results in Table 1?** N/A means that the original paper did not report results for that dataset; For example, ANT-A* did not test on CIFAR100. However, we produced the DNDF (no reported CIFAR results), classic DT, and oblique DT (using public OC1 from original authors) results ourselves. We will clarify this in the caption.
> - **In the “Compute node probabilities” paragraph, the definition of $r_i$ seems confusing.** We will clarify the definition of r_i below and also in the manuscript. First, we consider the k classes and their corresponding row vectors in the fully-connected layer’s weights. The jth class’s representative r_i is the jth row vector in the fully-connected layer’s weights. Second, for every inner node in the tree, we consider the subtree that node is the root of. Take all leaves in that subtree, and average all leaf r_i’s. That average is the inner node’s r_i.
> - **What does NN mean? How does it differ from CNN-RNN?** NN means the original Neural Network with standard training, without involving hierarchies. By contrast, CNN-RNN utilizes an RNN to generate sequential labels (the RNN rather than a fixed hierarchy generates the sequence of decisions). On a subset of ILSVRC 2010, Wang et al (2016) show significant improvement over a baseline neural network. We compare NBDT accuracy with both the NN and CNN-RNN in Table 1.

---

> ### Author Response · Authors · 2020-11-25
> **Manuscript Update**
>
> Thank you again for your feedback and suggestions. We have added the suggested clarifications; we have also improved Sec 3's clarity by incorporating responses to your clarification questions above.

---

### Official Review · AnonReviewer5 · 2020-11-08
**Take a step further on integrating the NN and Decision Tree**

**Rating:** 8
**Confidence:** 3

**Review:**

Aim to improve the interpretability and the accuracy of the neural network, this paper takes a step further on the integration of NN with a decision tree. It will replace the final linear layer of the NN with a decision tree induced by pre-trained model weights. It takes advantage of both hard and soft decision trees and designs suitable tree supervision loss thereon. Extensive experiments verify the design choice of the proposed components. On both small-scale and large-scale datasets, it beats the decision tree counterparts. Also, on the aspects of generalization and interpretability, it shows the strength compared to NN.

This work is a good try to combine the two techniques NN and decision tree. It finally makes the combination to achieve comparable accuracy with the NN and also enjoy the benefit in the aspects of generalization and interpretability. Recent SOTA of capsule networks which are based on the NN backbone and this work are both achieved comparable performance with NN. They show a promising direction for studying representation learning. Researchers can delve deeper based on this work to further exploit how to integrate decision tree into NN and the characteristics of the combination (e.g. adversarial examples).

With the decision tree, we can visualize the decision process the bring the benefits of interpretability. The paper proposes to label the decision nodes with WordNet and show the applications of zero-shot generalization, high-level concepts, dataset debugging, and improved human trust. There are lots to do on the aspects. Also, the zero-shot and high-level concept experiments are really intriguing. Using the pre-trained model weights to construct the tree and the proposed tree losses to train can help the generalization in such a significant way, though the performance would depend on the accuracy of the superclasses labeling and the agglomerative clustering. Where the benefits come from? The method is only used the same information as the NN and the tree is also constructed based on the pre-trained weights. Does the way of making hierarchy decisions help here? If you do not enforce the second term of the equation (3), will the phenomenon be the same?

Overall, the paper is very easy to follow and the figures really help understanding. Extensive experiments help to know the performance, effectiveness of the proposed components, and also its unusual applications.

----------------------------
Some concerns and comments are listed below:

Will you update the weights of the intermediate nodes?

On large-scale datasets, the paper currently only tests using the EfficientNet. The reviewer wonders if the author can use more advanced backbones to see the performance changes.

The reviewer is unsure of the specific way to label the decision nodes. Will you use the wordvec provided in the wordnet and compare it with the decision nodes' feature? Since your structure is different from the WordNet, how do you match the classes with the nodes?

-----------------after rebuttal------------

I would like to keep my origin score due to the pros listed above.

---

> ### Author Response · Authors · 2020-11-14
> **Intuition for Improvement, WordNet Assignment**
>
> **Where do the benefits come from? Does the hierarchy help?**
>
> We believe benefits come from explicitly learning higher-order concepts (like Animal, Vehicle). This is supported by the generalization results in Table 5, where NBDTs see disproportionate accuracy boost higher up in the hierarchy (9-16%); this is in stark contrast to the overall accuracy boost of 1-2%, suggesting higher-order decisions dominate the improvements.
>
> We agree that the hierarchy is both helpful and important.  We also show that the wrong choice of hierarchy can actually hurt accuracy (Table 2).
>
> **How do you match WordNet concepts with your hierarchy, since the tree structures differ? Do you use Word2Vec?**
>
> Although this is an interesting suggestion we will consider, we currently do not need Word2Vec.
>
> To assign WordNet meaning to nodes, we compute the earliest common ancestor for all leaves in a subtree: For example, say “Dog” and “Cat” are two leaves with a parent node. To find WordNet meaning for the parent, find all ancestor concepts that “Dog” and “Cat” share, like “Mammal”, “Animal”, and “Living Thing” (according to WordNet). The earliest shared ancestor is “Mammal,” so we assign “Mammal” to the parent node for “Dog” and “Cat”. We can repeat this for any intermediate node. In summary, for all leaves that are descendants of this intermediate node, find the earliest shared ancestor. We briefly mention this in Sec 5.1 and will expand that description.
>
> However, WordNet is missing many concepts that an intermediate node may represent: for example, the leaves may share context (e.g., “fish” and “boat” are both “aquatic”) or an attribute (e.g., “pencil” and “wire” are both cylindrical). We show examples of non-WordNet concepts in Appendix Figure 13. Note that these non-WordNet meanings require human intervention to annotate.
>
> **Clarifications**
> - **What happens if you enforce only the tree supervision loss (no cross entropy term)?** We found that, without the cross entropy term, the tree supervision loss alone does not boost accuracy. It seems that regular cross entropy is needed to first learn appropriate leaf representations before tree supervision loss can force the network to learn higher-order concepts like Animal. We will gladly include this ablation in Table 3.
> - **Do you update the weights of intermediate nodes?** Yes, but not directly. The intermediate nodes do not have weights themselves. Instead, the intermediate node weights are an average of its leaf weights (Sec 3.2). This means the intermediate node weights are updated when the leaf weights (i.e., fully-connected layer weights) are updated.
> - **Could you run the large scale datasets on more advanced backbones than EfficientNet?** We would be happy to test on more advanced backbones; did you have any in particular in mind? We chose EfficientNet (2019) simply because it had a) small enough models to fit in our compute budget and b) at the time, one of the latest open-sourced training pipelines.

---

> > ### Comment · AnonReviewer5 · 2020-11-24
> > **Response to authors**
> >
> > Thanks for your thorough answers. Your comments address my questions and fit my original thoughts and intuition. For the backbone, you can refer to [ImageNet Benchmark](https://paperswithcode.com/sota/image-classification-on-imagenet). Not need to test on the top one. Test on several different great backbones (e.g. ResNet and SENet) would help verify whether the method can achieve improvements in a general way.

---

> > > ### Author Response · Authors · 2020-11-25
> > > **Backbones, Manuscript Update**
> > >
> > > Thank you again for your feedback and suggestions. Although these new experiments won't make it in time for the rebuttal deadline, we will definitely consider the suggestion of validating NBDT improvements on common backbones like ResNet and SENet, on ImageNet. In the meantime, we have improved Sec 3's clarity by incorporated responses to your clarification questions above.

---

### Decision · Program_Chairs · 2021-01-07
**Final Decision**

**Decision:**

Accept (Poster)

**Comment:**

This paper initially received mixed ratings but after the rebuttal, all reviewers recommended acceptance. Reviewers appreciate the novel technical ideas and extensive experimental results.